# Separation of N–C$_5$H$_{12}$–C$_9$H$_{20}$ Paraffins Using Boehmite by Inverse Gas Chromatography

José L. Contreras-Larios [1], Antonia Infantes-Molina [2], Luís A. Negrete-Melo [1],
Juan M. Labadie-Suárez [3], Hernani T. Yee-Madeira [4], Miguel A. Autie-Pérez [3,*] and
Enrique Rodríguez-Castellón [2,*]

[1] Universidad Autónoma Metropolitana-Azcapozalco CBI, Energía, Av. Sn. Pablo 180 Col. Reynosa, C.P. 02200 México D.F., Mexico; jlcl@correo.azc.uam.mx (J.L.C.-L.); luis.alfonso.negrete@gmail.com (L.A.N.-M.)
[2] Departamento de Química Inorgánica, Cristalografía y Mineralogía, Facultad de Ciencias, Universidad de Málaga, Unidad Asociada al ICP/CSIC, 29071 Málaga, Spain; ainfantes@uma.es
[3] Facultad de Química, Universidad Tecnológica de la Habana, José Antonio Echeverría, 10400 Havana, Cuba; juanm@quimica.cujae.edu.cu
[4] Escuela Superior de Física y Matemáticas, Instituto Politécnico Nacional, C.P. 02200 México D.F., Mexico; hernanyyee@hotmail.com
[*] Correspondence: aautie@gmail.com (M.A.A.-P.); castellon@uma.es (E.R.-C.)

**Abstract:** The separation of a mixture of C5–C9 *n*-paraffins was achieved by Inverse Gas Chromatography (IGC) by using boehmite; AlO(OH), in a packed column with short exposure times and temperatures; from 45 °C to 52 °C. The boehmite was characterized by XRD; ATG; SEM; IR spectroscopy and N$_2$ adsorption. The material exhibited a low crystalline boehmite (AlOOH) structure and presented high hydration (pseudoboehmite). The reverse gas chromatography measurements showed that the elution temperatures of the C5–C9 *n*-paraffins were low compared with those obtained for other adsorbents. The differential heat of adsorption values ensures the satisfactory separation of the components in the C5–C9 mixture under suitable chromatographic conditions.

**Keywords:** n-C5–C9 separation; boehmite; inverse gas chromatography; differential heat of adsorption

## 1. Introduction

Bohemite [AlO(OH)] is an aluminum oxy-hydroxide present naturally in the bauxite ores, along with diaspore (AlOOH) and gibbsite [Al(OH)$_3$]. Pseudo-boehmite is a poorly crystallized boehmite [1,2], consisting of Al(O,OH)$_6$ octahedral layers, and used as a catalyst or catalyst support in several processes [3–5].

One of the most important applications of the boehmite is as precursor to obtain γ–Al$_2$O$_3$. After partial dehydroxylation from 300 to 550 °C, boehmite is converted to γ–Al$_2$O$_3$ [2,6]. Among the aluminas, γ–Al$_2$O$_3$ nd η–Al$_2$O$_3$ are important as catalyst supports, due to their highly effective contact area, high thermal stability, and surface acidity susceptible to be controlled.

It is possible to obtain the pseudoboehmite (PB) synthetically depending on the characteristics of the synthesis method employed. It has been reported the synthesis by a sol-gel method with and without catalyst [7], the atrane method [8] and low-cost approaches to synthesize PB with high surface area and broad pore size distribution. Moreover, previous reports have been focused on acid neutralization by using sodium aluminate as Al source [9], the hydrolysis of aluminum alkoxides [10] and a modified cation–anion double hydrolysis method by using water glass as the pore-expanding agent.

The aim of this work is to show the capability of this adsorbent to separate a paraffins mixture of n-C5–C9 using the inverse gas chromatography (IGC). The IGC is a technique for obtaining information

on the physico-chemical characteristics of the solid surface under flow conditions and its fundamentals are well explained in several publications [11,12]. Briefly, this technique is to investigate the ability of a porous material to separate a mixture of compounds by placing the porous solid in a packed column, which is crossed by the flow of the carrier gas and the mixture. At the end of the column, the mixture is analysed using a chromatographic detector. This technique is able to make the study at several carrier gas flows and temperatures. It provides information about the interactions between gases and vapors with solid surfaces as diverse as alumina [13], chromia [14,15], polymer fabrics [16], biopolymers [17], nanomaterials [12], hexacyanocobaltates [18], zeolites [19]. The possible use of a Cuban mineral, in its natural and modified state, for the separation of hydrocarbons components and C5–C9 *n*-paraffin mixture has been studied too [20,21]. It has been used in phenolic resin-carbon molecular sieve membranes for $C_3H_6/C_3H_8$ gas separation [22] and zeolite membranes [23].

Boehmite has been used to obtain a nanostructured alumina with excellent mechanical crush strength [24], to prepare spherical silicalite/$\gamma$-$Al_2O_3$ granules with high crush strength and good sorption properties for $SO_2$ [25]. And, to the best of our knowledge, no references about the use of synthetic bohemite in the adsorption and separation of hydrocarbons mixtures were found. For these reasons, the aim of this work was to study the behavior of a synthetic boehmite in the adsorption and separation of C5–C9 paraffin mixture by IGC. Moreover, it is going to be used the Dubinin's two term equation to describe the performance of gas adsorption in this adsorbent, which has not been described yet.

## 2. Materials and Methods

### 2.1. Preparation of Boehmite

The synthesis of bohemite was carried out by using the co-precipitation method. To this end, stoichiometric amounts of $Al_2(SO_4)_3$ and aqueous $NH_3$ were added to distilled water in a stainless steel reactor under vigorous stirring. In a typical synthesis, 15 g of $Al_2(SO_4)_3$ were added to 240 mL of water. Once dissolved, the solution of aqueous $NH_3$ (50 wt%) was added dropwise under stirring to reach a pH value of 8. At this stage, it was observed the formation of a white gel, which was aged at 60 °C under stirring for 24 h. The white precipitate formed was recovered by filtration, washed several times with distilled water and dried in an oven at 110 °C for 24 h.

### 2.2. Characterization Techniques

In order to make the analysis of X-ray diffraction, a Phillips diffractometer (X'Pert) was operated at 30 kV and 20 mA using a Cu-K$\alpha$1 radiation ($\lambda$ = 0.1541 nm) and a rotation speed of 0.02°/min, the angle 2$\theta$ changed between 20° and 80°.

Thermogravimetric analysis (TG) was made by loading 10 mg of boehmite in a horizontal balance with total capacity of 200 mg, with a sensitivity of 0.1 $\mu$g, (SDTQ-600 instrument) under a $N_2$ atmosphere (Infra-Air Products, 99.9%) and a flow rate of 100 $cm^3$/min. The sample was introduced into the instrument without pre-treatment and the heating rate was 10 °C/min. The temperature increased from 25 to 1000 °C. All the weight changes were recorded as a function of temperature and the derivate (dW/dT) (DTG) was calculated in order to know the temperatures related with the inflexion points.

The infrared spectra were obtained by using samples pulverized with KBr in a mortar and pressed up to 700 kgf/$cm^2$ to obtain a transparent disk in order to be placed in the infrared beam path of the instrument. The infrared spectrometer used was a Varian Spectra 220.

Scanning electron microscopy was carried out with a scanning electron microscope FIB Mod. FEI Quanta 3D FEG Co. The qualitative chemical analysis and the corresponding spectra were obtained by EDS (X-ray energy dispersive spectroscopy) with a probe coupled to SEM microscope. The powder samples were spread on a graphite tape with adhesive on both sides, and then a mixture of Au and Pd atoms was evaporated on the samples to make them conductive.

The specific surface area was determined by $N_2$ physisorption in an ASAP-2000 instrument (Micromeritics). A sample of 500 mg was degassed at low pressure ($1 \times 10^{-3}$ Torr) and at 300 °C for 1 h. The sample was then cooled at −196 °C and the $N_2$ physisorption was made. The isotherm was determined and the surface area, pore volume and pore diameter were calculated from the BET method.

For a more detailed analysis of the $N_2$ adsorption isotherm, the two terms of the Dubinin equation [26,27] were fitted using the following equation:

$$N_m = N_{m01} \, exp \, (-A/\beta E_{01})^2 + N_{m02} \, exp \, (-A/\beta E_{02})^2 \tag{1}$$

where $N_{m01}$, $N_{m02}$, $E_{01}$, $E_{02}$ are the maximum adsorption values in mmol/g and the characteristic adsorption energies in J/mol in the narrow and widest pores respectively, β is the affinity coefficient and A is the adsorption potential defined as:

$$A = RT \, ln \, (Po/Pe) \tag{2}$$

where *Po* is the vapor pressure at experimental temperature, *Pe* is the equilibrium pressure and *R* is the universal gas constant.

Dubinin´s two-term equation efficiently describes the performance of gas adsorption in adsorbents with two pore distributions as carbons [5,6] and it was used in zeolites too [19,28], but it was not reported his use in boehmite.

### 2.3. IGC Determinations

The adsorption and separation studies of *n*-paraffin mixtures were carried out by IGC. For this purpose, a Shimadzu chromatograph, model 14B, equipped with a flame ionization detector (FID) was used in a temperature range of 22–170 °C. Helium was used as a carrier gas using a flow rate of 1–5 $cm^3$/min. The column, stainless steel with a length of 90 cm and an internal diameter of 0.3 cm, was packed with 5.0 g (6 $cm^3$) of boehmite sample. The column was conditioned overnight at 150 °C under helium flow. The *n*-alkane hydrocarbons (analytical quality) in their vapor phases were injected in the smallest detectable quantity, in order to meet the requirement of adsorption to infinite dilution, which corresponds to zero coverage. To calculate the adsorbate retention time ($t_R$), the retention time ($t_0$) of the lower adsorbent marker (methane) was used as a reference. The corrected retention times were taken as the average value of three injections for each adsorbate. The net adsorbate retention volumes (Vn) were calculated from the Equation (3):

$$Vn = JF(t_R - t_0)(T_C/T_A) \cdot (P_0 - P_W)/P_0 \tag{3}$$

where *J* is the James-Martin gas compressibility correction factor, *F* is the carrier gas flow rate at the flowmeter temperature (*Tf*), $T_C$ is the column temperature, $T_A$ is the room temperature ($T_A = Tf$), $P_0$ is the outlet column pressure, and *Pw* is the vapor pressure of water at *Tf*. The differential heat of adsorption ($Q_{dif}$) was obtained from chromatographic measurements at infinite dilution (Henry zone) using the following expression and the corresponding linear dependence ln $K_S$ vs. $1/T_C$:

$$Q_{dif} = -R[d(lnK_S)/d(1000/Tc)] \tag{4}$$

where R is the universal gas constant, $K_S$ is the surface partition coefficient for solute ($K_S = Vn/m \cdot Ss$), defined as the net retention volume per unit of sample mass in the column (m) and the specific surface area of the adsorbent (*Ss*).

The separation of the *n*-paraffin mixtures ($C_5H_{12}$-$C_9H_{20}$) was evaluated by the Bering and Serpinski's separation coefficient ($K_{BS}$). The temperature dependence of this coefficient for a mixture of two species is given by the following equation:

$$ln \, K_{BS} = (Q_{dif2} - Q_{dif1})/RT \tag{5}$$

where $Q_{dif1}$ and $Q_{dif2}$ are the differential heats of adsorption of the species 1 and 2 and $T$ is the column temperature. When $K_{BS} > 1$ the total species separation is possible.

## 3. Results

### 3.1. Samples Characterization

The X-ray diffractogram showed broad peaks (Figure 1) typical of a boehmite structure of low crystallinity or highly hydrated boehmite [29] verifying that the material synthesized was as expected. Thus reflections of the prepared sample appeared at 2theta (°) = 14, 28, 38, 49, 65, 70 and 85, and they were assigned to an orthorhombic $\gamma$–AlOOH. Compared with the standard diffraction peaks (JCPDS Card No. 21–1307), no other peaks were observed.

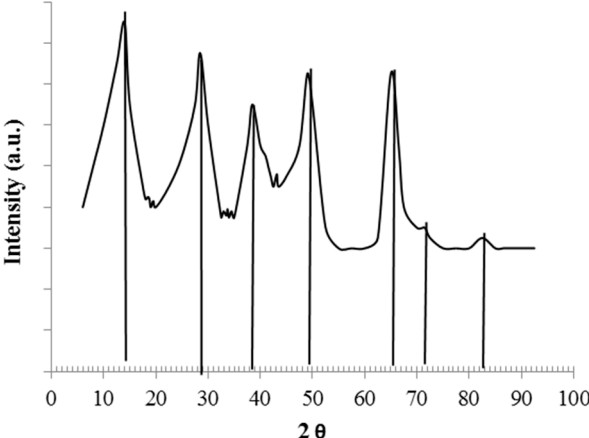

**Figure 1.** XRD of pseudoboehmite.

The thermogravimetric analysis (TGA) of the synthesized material is represented in Figure 2. The TG curve (blue line) showed a continuous weight loss and according to some research in the literature [30], the TG temperature range has been divided into four regions: Zone I (25–180 °C) was attributed to the loss of physisorbed water and there may be a loss of physisorbed gases. Zone II was associated with the decomposition of boehmite forming water (180–400 °C). Some studies have found combustion peaks attributed to surfactants in this temperature range [30]. Zone III (400–730 °C) was attributed to the elimination of residual carbon compounds and sulfur oxides that remained after the preparation of alumina with $Al_2(SO_4)_3$ and the resulting water release of the phase that changes to $\gamma$–$Al_2O_3$ [31]. Specifically, this change was between 370 and 630 °C, and is produced by dehydroxylation and transformation of boehmite to $\gamma$–$Al_2O_3$ [32]. In zone IV (750–1000 °C) the typical phase changes of $\gamma \rightarrow \delta$, and $\theta \rightarrow \alpha$ $Al_2O_3$ took place. The DTG curve evidenced several peaks associated to these weight losses. At low temperature, two peaks at 50 and 80 °C were present and correspond to evaporation of adsorbed water and or physisorbed gases. In the second-third temperature range the two observed peaks were attributed to the dehydroxylation of boehmite to form $\gamma$–$Al_2O_3$ and finally, the fourth peak was associated to phase changes [33].

These results indicated that the activation of the material synthesized inside the chromatographic column is possible at temperatures below 200 °C without phase transformation.

Considering the Infrared spectrum of boehmite (Figure 3a), in the OH stretching region it is evidenced that, besides the quite sharp bands assigned to surface hydroxyl groups (3800–3500 cm$^{-1}$), a broad absorption was present in the region 3500–2500 cm$^{-1}$ and assigned to H-bonded groups [34].

Moreover, the sharp band near 1078–1150 cm$^{-1}$ was due to the asymmetric stretching of the internal $AlO_4$ tetrahedra; and the two contributions at about 900 and 700 cm$^{-1}$ were attributed to symmetric bending of O-Al-O groups. The water of hydration produced a band at 1550–1800 cm$^{-1}$ region, present in both samples [34]. These vibrations were also present in the sample of $\gamma$–$Al_2O_3$

(Figure 3b), although slightly more intense. Only the band at 700 cm$^{-1}$, belonged to Al = O groups in boehmite [35], was less intense. It was observed a small band in 2400 cm$^{-1}$, apparently caused by molecular $CO_2$. If both spectra are compared, significant differences in the intensity of the bands are found. The intensities of all peaks were strongest for $\gamma$–$Al_2O_3$.

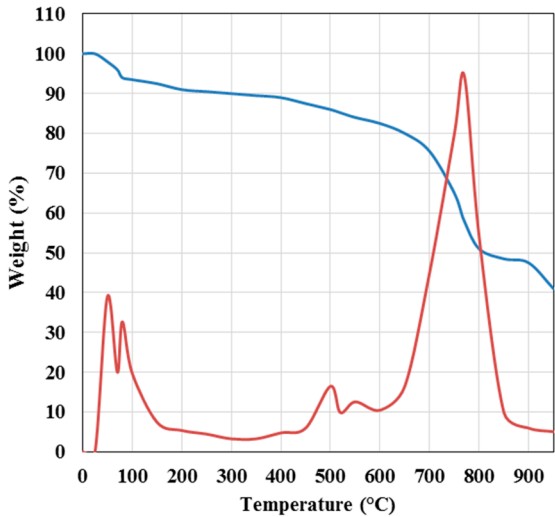

**Figure 2.** TG (Blue line) and DTG (Red line) of boehmite.

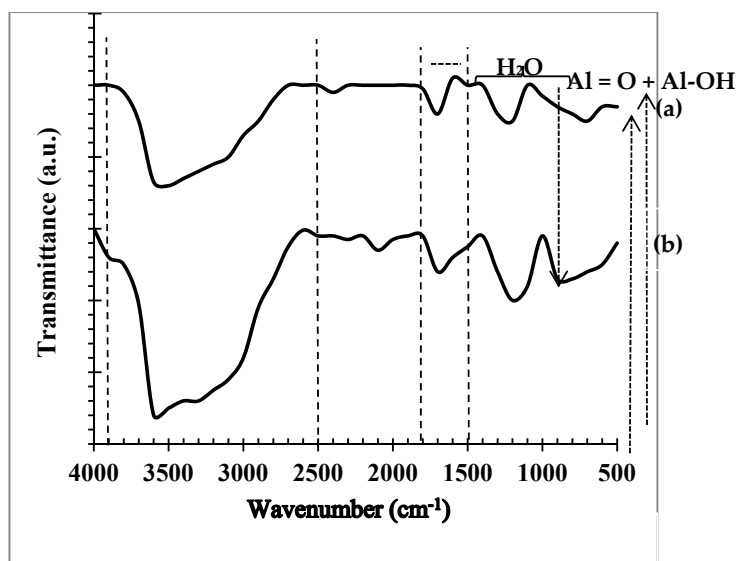

**Figure 3.** IR spectra of boehmite (**a**) and $\gamma$–$Al_2O_3$ calcined at 550 °C (**b**).

Information about the morphology was obtained from SEM microscopy. Various morphologies of boehmite nanostructures were observed, such as nanosheets (Figure 4a) and nanoparticles (Figure 4b) with flowerlike three-dimensional nano-architectures (Figure 4c). The nanosheet morphologies of boehmite have been seen in other investigations [35]. Nanoparticles were also observed by Naskar et al. [36]. Finally, flowerlike three-dimensional nanoarchitectures of boehmite have been observed by Zhang et al. [37].

This preparation of boehmite showed nanosheets assembled with nanoparticles in the absence of surfactants at a lower reaction temperature, such as 60 °C. Other authors have found nanosheets when they prepared boehmite samples at a pH near 7 [38]. The porosity properties produced by the obtained boehmite nanosheets demonstrated that the material can be utilized as adsorbent to separate the paraffin mixture.

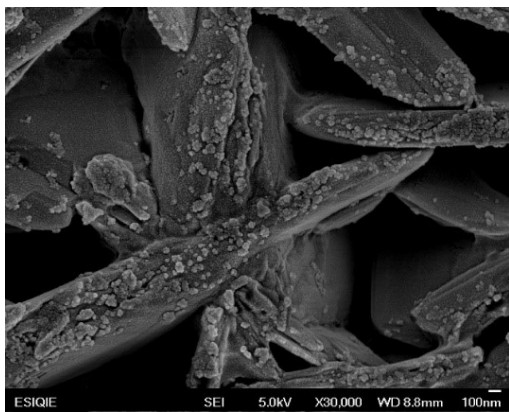
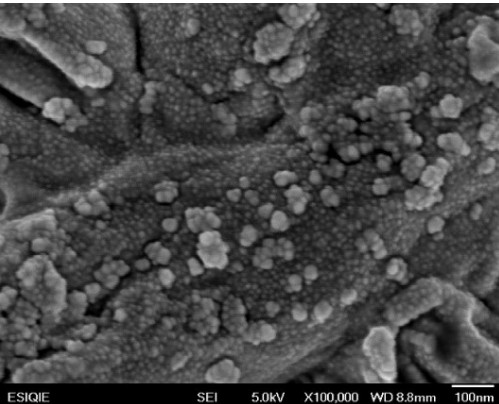

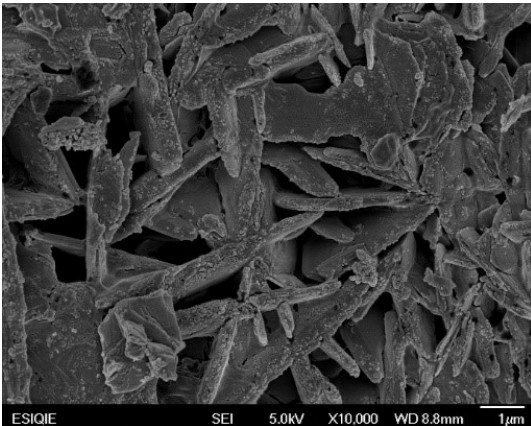

**Figure 4.** SEM micrographs of the as-prepared boehmite

The $N_2$ adsorption isotherm at −196 °C was, according to the IUPAC classification, of type III, typical of microporous solids, and showed a nearly linear dependence between the adsorbed amount of $N_2$ and the equilibrium pressure up to approximately 475 mmHg. After this pressure, it was clearly distinguished a gradual slope and a final sharp rise that is characteristic of capillary condensation (Figure 5A).

The specific surface area (Se) of 10.7 $m^2$/g, obtained by the BET method, showed that during the process of synthesis the clumping of small pseudo-boehmite particles should occur. This could be explained due to the great speed of hydrolysis of aluminum precursor that can originate different morphologies and sizes of the synthesized material [39], thereby promoting bonding of particles to reduce the surface energy of the system. Additionally, it is known that the diffusion of $N_2$ molecules at −196 °C in the micropores of a material is difficult and therefore discarding at that temperature the diffusion through the narrow spaces between the layers of boehmite particles linked by -OH groups. This was confirmed when the curve of dependency of the amount of adsorbed $N_2$ as a function of time was analyzed (Figure 5B) and compared to that of dealuminated clinoptilolite [19]. In the latter case, $N_2$ was disseminated inside the solid and the adsorption process lasted more than three hours, whereas boehmite $N_2$ adsorption hardly lasted three minutes. Moreover, on the same curve it was observed that the adsorption process occurred in three stages: the first to 47 mmHg (A-B), the second between 59 and 519 mmHg (C-D) and third corresponding to abrupt capillary condensation, from the equilibrium pressure of 537 mmHg (E-F).

The isotherm range before the capillary condensation made by the two terms of Dubinin method showed the existence of two zones in the mesoporous system. The Dubinin-Stoeckli criterion [27] was applied to determine the average pore diameter (Dp = /Ec) and a narrow pore region centred at Dp = 5.2 nm was found as well as another region with mean pore widths of Dp = 19.2 nm, being the last pore diameters large enough to allow diffusion of big molecules (Table 1).

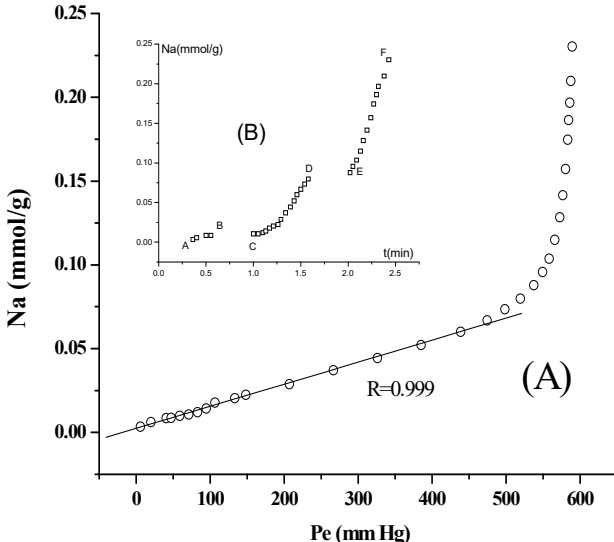

**Figure 5.** (**A**) Nitrogen adsorption isotherm in boehmite at −196 °C; (**B**) Nitrogen adsorption isotherm in boehmite as a function of time at −196 °C.

**Table 1.** Parameter settings of two terms Dubinin equation obtained from nitrogen adsorption isotherm.

| Zone | $N_m$ [mmol/g] | $E_c$ [J/mol] | $D_p$ [nm]. |
|------|------------|-----------|----------|
| 1 | 0.041 | 2481 | 5.2 |
| 2 | 0.056 | 669 | 19.4 |

Nm: Maximum adsorption Dubinin areas. Ec: characteristic energies. Dp: and average pore diameter.

It is believed that the fast $N_2$ adsorption at −196 °C occurs on the mesoporous surface formed during synthesis and it is observed a satisfactory correspondence between the fitted isotherm and the experimental isotherm as well as the kinetics (Figure 6A,B). In both isotherms, it was shown that the adsorption during the first stage of adsorption essentially corresponded to the first area of Dubinin mesopores and the adsorption during the second stage occurred in the second zone of Dubinin mesopores, respectively.

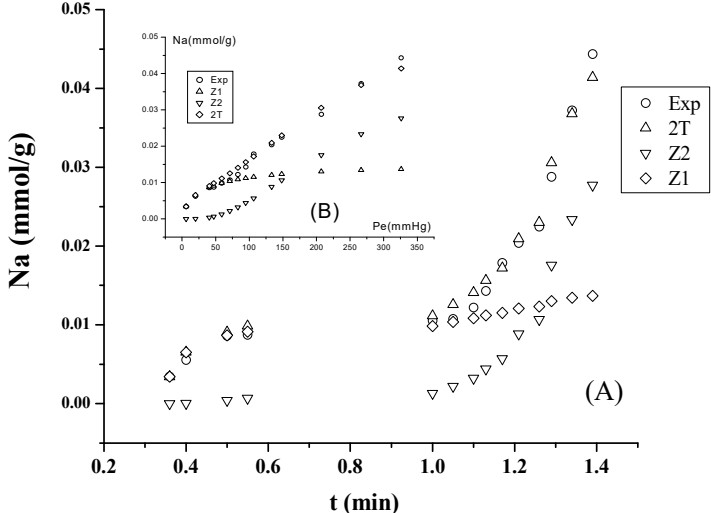

**Figure 6.** Experimental and fitted adsorption isotherms of adsorbed nitrogen in Bohemite at −196 °C: (**A**) as a function of time and (**B**) as a function of pressure until the equilibrium pressure of 325 mmHg (Experimental Isotherm (Exp), setting of Dubinin Zone 1 (Z1), setting of Dubinin Zone 2 (Z2) and setting for two terms of Dubinin (2T)).

### 3.2. Separation of n-Paraffins C5–C9 by IGC

From the results of $N_2$ adsorption seen above, we can infer that: (i) larger molecules than $N_2$, as is the case of *n*-paraffins (9C9), can be adsorbed and diffused into the material mesoporosity and (ii) the differences in dispersion interactions between these molecules and the solid surface must be due to the quantity of -$CH_2$- groups in the *n*-paraffins chain.

An important experimental feature is the low column temperature necessary to separate *n*-paraffins (C5–C9). As reported in previous studies, the separation of *n*-C3-*n*-C7 mixtures in hexacyanocobaltates [18], *n*-C3-*n*-C9 mixtures in clinoptilolite [19] and *n*-paraffins and aromatics in zeolites, elution temperatures above 127 °C were required [40]. Instead, the separation of *n*-C5-*n*-C9 mixtures in a larger boehmite column (90 cm) was achieved at elution temperatures below 77 °C. These data were a first indication that the non-specific interactions between *n*-paraffins and boehmite were lower than those present in the other two adsorbents. Therefore, in the present study it was evaluated the separation of *n*-C5-*n*-C9 mixtures on synthetic pseudoboehmite. For the five studied *n*-paraffins, it was observed that the chromatographic peaks were sufficiently symmetric to verify that the experiments were performed to infinite dilution (Henry Zone), (Figure 7). Moreover, all the paraffins eluted at temperatures lower than 70 °C.

In all cases, the graphs of lnKs vs. 1000/Tc were about straight lines with R > 0.99 (Figure 8) which allow determination of the differential adsorption heats Qd from the slopes of these lines (Table 1). The Qd dependence as a function of carbon atom number of the studied *n*-paraffins was also a straight line (Figure 9) whose slope corresponds to the contribution of each -$CH_2$- group being 3.22 kJ/mol.

By comparing the obtained differential adsorption heat (Qd) values of the *n*-paraffins in boehmite with those obtained with the same *n*-paraffins (*n*-C5–C9) and other adsorbents, the values obtained in the present research are lower. Thus, in dealuminated clinoptilolite, Qd were in the range 33–53 kJ/mol [19] and in hexacyanocobaltates greater than 40 kJ/mol [18]. It is believed that the path followed by the molecules of *n*-paraffins during diffusion in these adsorbents is the cause of this great change in adsorption heats values. The molecules are diffused into the channels and cavities of dealuminated clinoptilolite and hexacyanocobaltates interacting in all directions, while in boehmite, the *n*-paraffins can only follow the outside of the particles, interacting with the outer surface, due to the layered structure of boehmite, where each layer consists of double chains of AlO(OH) octahedra linked by OH groups inside. The space between the layers of octahedra in boehmite has been reported to vary between 2.71 Å [41] and 2.50 Å [42]. The differences in size might be caused by water adsorbed between the layers [43]. However, there is a consensus that the spacing between layers is very small, due to the -OH groups that form strong hydrogen bonds holding together the layers, thus preventing the passage of molecules of *n*-paraffins C5–C9. Moreover, considering the cross-sectional area of analytical -$CH_2$- group (assuming the circular area) is about 6 Å$^2$, the *n*-paraffins C5–C9 can only diffuse through the mesopores and the exterior of the particles and interact with those surfaces and their -OH groups.

In this way, it becomes known that in spite of the relatively low Qd values, the differences between them are sufficient according to the Bering-Sierpinski ($K_{BS}$) separation criterion to perform the separation of the mixture *n*-C5-*n*-C9 satisfactorily (Figure 9), indicating that to achieve a proper separation of compounds, the Qd values are not as important as the differences between them.

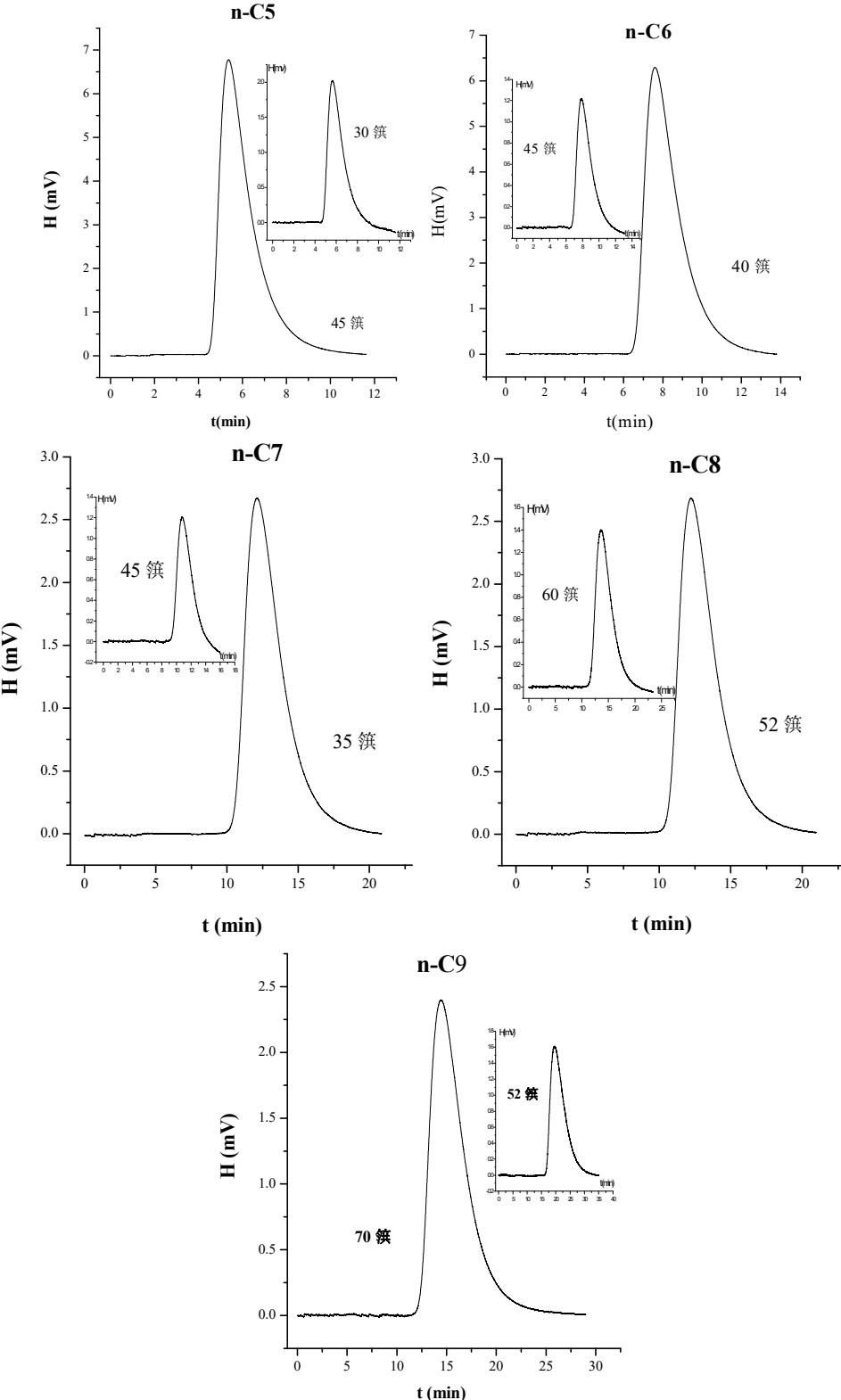

**Figure 7.** Chromatograms of *n*-C*n* in boehmite.

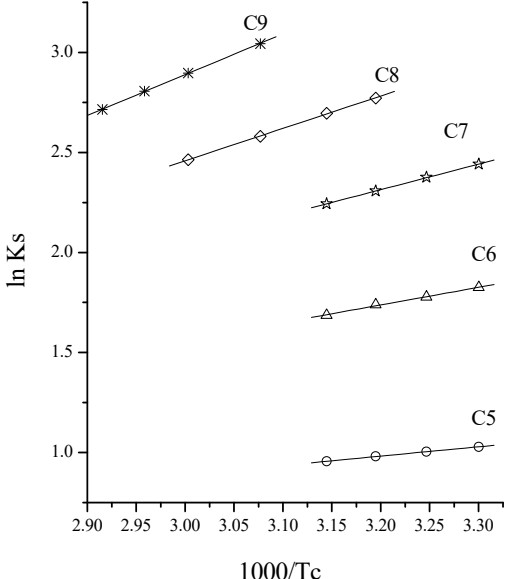

**Figure 8.** (ln Ks) versus (1000/Tc) for *n*-C$_5$-*n*C$_9$ mixture in boehmite for the temperature range of 30–70 °C.

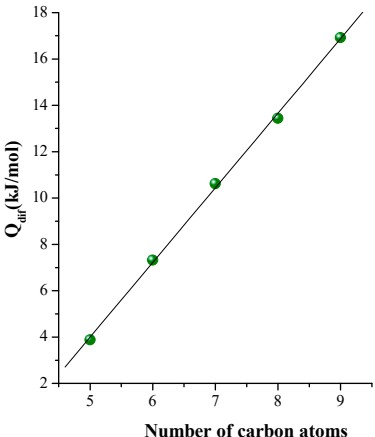

**Figure 9.** Dependence of differential adsorption heats (Q$_d$) on the number of carbon atoms C5–C9 in the *n*-paraffins.

## 4. Conclusions

A mixture of *n*-paraffins C5–C9 was separated using a synthetic boehmite as adsorbent. Unlike other adsorbents, the separation was possible at lower temperatures. All detectable peaks in the XRD pattern of boehmite were assigned to an orthorhombic structure of γ-AlO(OH) and the vibrations of Al = O and Al-OH were found by IR bands. This boehmite showed low values of N$_2$ adsorption and surface area (10.7 m$^2$/g) presenting small nanoparticles, nanosheets, and flowerlike structures as observed by SEM. The differential adsorption heats were much lower than those obtained with other adsorbents, but the differences between them were sufficient to ensure the separation of the *n*-C$_5$ to *n*-C$_9$ mixture.

**Author Contributions:** Conceptualization and Methodology, J.L.C.-L. and M.A.A.-P. Synthesis, L.A.N.-M.; Characterization, J.L.C.-L., H.T.Y.-M., A.I.-M. and E.R.-C. IGC tests, M.A.A.-P.; Writing, M.A.A.-P., A.I.-M., J.L.C.-L. and E.R.-C.

**Funding:** This research was funded by CONACYT (Mexico) NAME OF FUNDER and Ministry of Economy and Competitiveness of Spain, project CTQ2015–68951-C3-3-R, FEDER funds and Ramón y Cajal contract (RyC2015-17870).

**Acknowledgments:** Miguel A. Autie-Perez thanks CONACYT for funding in his stay in Mexico as a guest researcher in ESFM-IPN and in the Catalysis Laboratory of the UAM-Azcapotzalco, and University of Malaga for his stay in the Department of Inorganic Chemistry. This research was also financed by Ministry of Science, Innovation and Universities of Spain, project RTI2018-099668-B-C22 and FEDER funds. A.I.M. thanks the Ministry of Economy and Competitiveness for a Ramón y Cajal contract (RyC2015-17870).

**Conflicts of Interest:** The authors declare no conflict of interest.

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
