# Peer review of "Separation of N–C5H12–C9H20 Paraffins Using Boehmite by Inverse Gas Chromatography"

_applsci, doi:10.3390/app9091810_

Round 1
Reviewer 1 Report
This paper essentially describes the characterization of a particular form of alumina. The methods appear to be sound and the conclusions valid. There are a number of minor grammatical errors that will be fixed in proofing I think, for example:
Line 55: should read "studied" not "study".
Line 56: Should read "used in"
Line 81: should read "sensitivity" not "sensibility"
Line 112: should read "two-term" not "two terms"
114: should read "two pore" not "two pores"
Line 154: "the expected" should read "as expected".
Line 169: subscripting on numbers needed.
The Term "ATD" only appears in the figure 2 cation, needs to be defined somewhere.
Author Response
Thank you for your comments and suggestions
Line 55: should read "studied" not "study".
Done
Line 56: Should read "used in"
Done
Line 81: should read "sensitivity" not "sensibility"
Done
Line 112: should read "two-term" not "two terms"
Done
Line 114: should read "two pore" not "two pores"
Done
Line 154: "the expected" should read "as expected".
Done
Line 169: subscripting on numbers needed.
Done
The Term "ATD" only appears in the figure 2 cation, needs to be defined somewhere.
Thank you for this comment. The term "ATD" is wrong. The right is DTG.
Other grammar and misspelling mistakes were mended and indicated in the revised version.
Reviewer 2 Report
The paper entitled “Separation of n-C5H12-C9H20 Paraffins using Boehmite by Inverse Gas Chromatography” by José L. Contreras-Larios, Antonia Infantes-Molina, Luís A. Negrete-Melo, Juan M. Labadie-Suárez, Hernani T. Yee-Madeira, Miguel A. Autie-Pérez, Enrique Rodríguez-Castellón is trial to use chromatography to achieve an application of separation.
The format and English style is OK, except for the references where include several mistakes: Catal. Letters instead of Catal. Lett.
The paper deserves publication even though the insight is not that meaningful, or this could be solved, because as it is, the application seems poor.
Author Response
Thank for your comments and sugegstions.
We have corrected the mistakes indicated and the English has been adited again.